# PufferLib: Making Reinforcement Learning Libraries and Environments Play Nice

**Joseph Suarez**

## Abstract

Reinforcement learning (RL) frameworks often falter in complex environments due to inherent simplifying assumptions. This gap necessitates labor-intensive and error-prone intermediate conversion layers, limiting the applicability of RL as a whole. To address this challenge, we introduce PufferLib, a novel middleware solution. PufferLib transforms complex environments into a broadly compatible, vectorized format, eliminating the need for bespoke conversion layers and enabling more rigorous testing. Users interact with PufferLib through concise bindings, significantly reducing the technical overhead. We release PufferLib's complete source code under the MIT license, a pip module, a containerized setup, comprehensive documentation, and example integrations. We also maintain a community Discord channel to facilitate support and discussion.

## 1 Background and Introduction

Reinforcement Learning (RL) generates data through interaction with a multitude of parallel environment simulations. This dynamism introduces non-stationarity into the optimization process, necessitating algorithmic treatments distinct from those employed in supervised learning. When compounded by sparse reward signals, this issue yields several complications, including extreme sensitivity to hyperparameters, which extends even to the random seed. Consequently, experiments often yield unpredictable learning curves with spikes, plateaus, or crashes, deviating from the more reliable behavior observed in other machine learning domains.

Alongside this lies the pragmatic challenge of implementing RL in complex environments with currently available tools. Although this is arguably a more solvable problem than optimizing the online learning process, the lack of effective tooling often exacerbates the problem, making it an arduous task to resolve despite thorough investigation. These issues frequently cause significant delays, frustration, and stagnation in the field, potentially deterring talented researchers from pursuing work in this area.

In response, we introduce PufferLib, a novel middleware solution bridging complex environments and reinforcement learning libraries, effectively mitigating the integration challenges. PufferLib decouples one layer of RL's unique complexities, making the remaining challenges more manageable and fostering more rapid progress in the field. Other existing solutions such as Gym [Brockman et al., 2016], PettingZoo [Terry et al., 2020b], and SuperSuit [Terry et al., 2020a] aim to define standard APIs for environments and implement common wrappers. PufferLib builds on Gym and PettingZoo but also addresses their specific limitations, which we will discuss after providing comprehensive context for the problem at hand.

PufferLib allows users to wrap most new environments in a single line of code for use with popular reinforcement learning libraries, such as CleanRL [Huang et al., 2021a] and RLlib [Liang et al.,

Submitted to the 37th Conference on Neural Information Processing Systems (NeurIPS 2023) Track on Datasets and Benchmarks. Do not distribute.

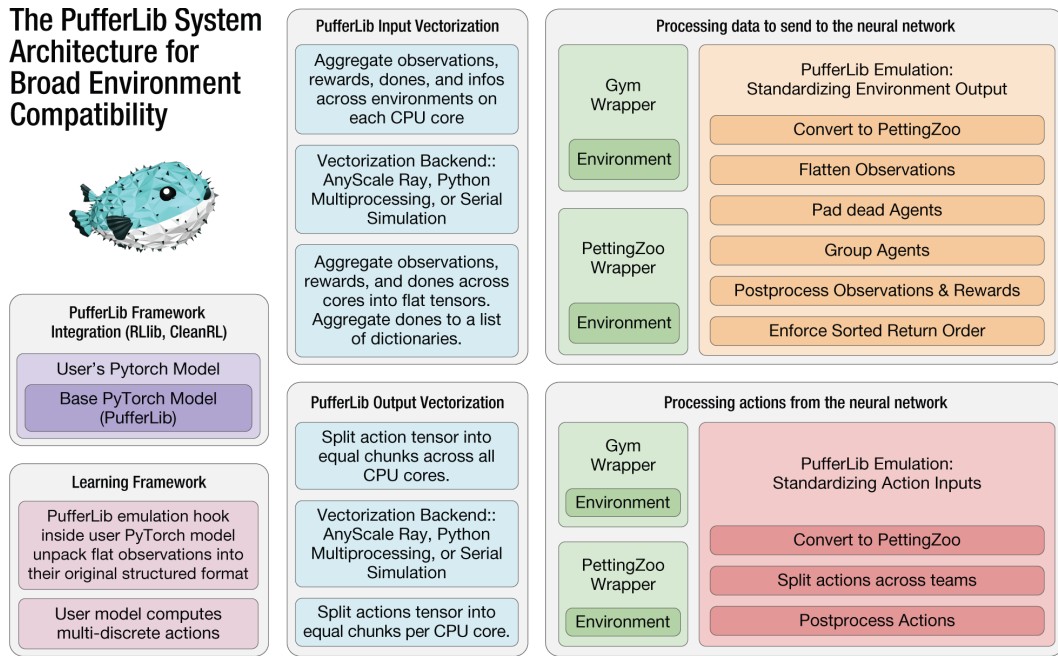

Figure 1: Detailed but non-comprehensive illustration of the PufferLib system architecture, comprising emulation, vectorization, and learning framework integrations. The orange emulation block demonstrate how PufferLib receives and processes environment data. The red emulation block demonstrates how PufferLib processes actions from the neural network to send to the environment. The blue vectorization blocks aggregate and split data received from and sent to the environment. Finally, the pink and purple blocks summarize how PufferLib provides compatibility with multiple frameworks given a single PyTorch network.

2017]. It natively supports multi-agent and variable-agent environments and addresses common complexities that include batching structured observations and actions, featurizing observations, shaping rewards, and grouping agents into teams. PufferLib is also designed for extensibility and is capable of supporting new learning libraries with a complete feature set in typically about a hundred lines of code.

## 2 Problem Statement

To thoroughly ground our work, we will walk through the intricacies of the transformations that reinforcement learning data must undergo, and demonstrate the shortcomings of existing approaches. Specifically, we will trace the required transformations from simulation onset to data processing by the initial model layer, and from action computation to the point when those actions influence the environment.

We will use Neural MMO [Suarez et al., 2021], a Gym and PettingZoo-compliant environment, as our guiding example. This environment, encapsulating many complexities common to advanced environments, features 128 agents competing to complete tasks in a procedurally-generated world. It provides agents with rich, structured observations of their surroundings and a hierarchical action space for interactions.

The environment initialization starts with a configuration file and a reset to yield an initial set of observations. This results in a dictionary of 128 individual observations, each of which is a structured dictionary housing differently-shaped tensors related to various aspects of the observation. As a part of the environment's standard training setting, these agents are grouped into teams of 8. Each team observation is then processed by a featurizer to yield a single structured observation, aggregating

information from across the team's agents. Subsequently, this observation must be batched for model usage.

This introduces two challenges. Firstly, since the observation is structured, we cannot merely concatenate tensors; we must concatenate each sub-observation across agents. Secondly, many learning libraries presuppose that observations can be stored in flat tensors, thus requiring data flattening. Following this, the data must be concatenated with information from several parallel environment instances. Once done, the data can be forwarded to the network.

We now encounter another problem: the network itself is structured, and attempting to learn from the flattened representation is akin to unraveling an image and using dense layers. Therefore, the structured observation representation must be recovered in a batched form, allowing for efficient processing of each sub-observation across all teams and environments in parallel. The model then computes a multidiscrete output distribution and samples an integer array for each team and environment. The output data is divided across environments, and each multidiscrete action is mapped into a structured format where each integer signifies a specific agent's action within a team. Finally, the environment can execute its first step.

Regrettably, this is the least complex step. All preceding actions must be reiterated, but with additional complexities. For example, the environment must now also return *rewards*, *dones*, and *infos*. These outputs, particularly rewards and *dones*, require grouping by team. For each team, we must track which agents have completed their tasks and signal that team is *done* only when all agents have finished. Similarly, we need a method to post-process and group *reward* signals per team. Since most learning libraries anticipate each agent to return an observation on every step, we must zero-pad the tensor for any agents that are *done*. Moreover, as the PettingZoo API does not mandate a consistent observation return order (a common source of bugs), we must verify this as well.

As illustrated, considerable work is needed to ensure compatibility between the environment and standard learning libraries - even for a fully Gym and PettingZoo-compliant environment like Neural MMO. We have provided support to the Neural MMO team in integrating PufferLib, and prior to integration, about a quarter of the Neural MMO code base was devoted to these transformations. This was also the primary source of bugs, many of which would lead to silent performance degradations. For instance, specific patterns of agent deaths could cause incorrectly ordered observations, leading to neural networks optimizing trajectories assembled from different agents. In another case, a bug in the reconstruction of observations misaligned data, causing incorrect subnetwork processing. Despite a strong engineering focus on testing, these bugs are two among dozens that reportedly emerged during Neural MMO's development.

## 3 Related Tools

Gym and PettingZoo, the prevalent environment APIs for single-agent and multi-agent environments respectively, offer several tools to mitigate the complexities described earlier. Supplementary third-party tools, like SuperSuit, provide standalone wrappers, while numerous reinforcement learning libraries furnish wrappers compatible with their internal APIs. For instance, Gym provides a range of wrappers for image observation preprocessing, observation flattening, action and reward postprocessing, and even sanity check wrappers for bug prevention. SuperSuit further adds multi-agent wrappers specifically designed to address the agent termination and padding issues discussed previously.

Current methodologies present some significant challenges. The tools described are designed as a set of wrappers applied sequentially to an environment instance, implying that (with a few exceptions), they should function in any order. However, particularly with PettingZoo, which caters to multi- and variable-agent environments, the gamut of possible environments is vast and challenging to test. This often results in scenarios where a bug in one wrapper causes an error in a different wrapper. Identifying the origin of such errors across multiple wrapper classes can be an overwhelming task, contributing to a general sense of frustration common in reinforcement learning research.

Moreover, the coverage of wrappers is insufficient. Despite the difficulties in testing and maintaining compatibility among existing wrappers, more are still needed. As it stands, there is no wrapper ensuring consistent agent key ordering, despite many reinforcement learning libraries demanding this. No wrapper exists for grouping agents into teams, a common operation, nor a wrapper that inherently vectorizes multi-agent environments across multiple cores. The current workarounds for the latter are unstable, abusing single-agent vectorization code. While additional development could resolve these issues, it would further aggravate the existing compatibility problem.

Another challenge is that some wrappers are infeasible to construct using the above approach. An observation unflattening wrapper, often needed to store observations in flat tensors while retaining the structured format for the model, is one such example. If the flattening wrapper is not the outermost one, the observation space structure required to unflatten the observation is lost. Conversely, if the flattening wrapper is always the final layer, all other wrappers must handle structured observation spaces, thereby adding unnecessary complexity and error-prone code.

## 4 PufferLib's Approach

PufferLib aims to handle all the complex data transformations discussed above, returning data in a format compatible with even the most basic reinforcement learning libraries. The system comprises three primary layers: emulation, vectorization, and framework integrations. The ultimate outcome allows users to write one-line bindings for some of the most intricate reinforcement learning environments available and use a single PyTorch network to train with multiple reinforcement learning frameworks.

### 4.1 Emulation

This layer forms the core of PufferLib. By applying the aforementioned data transformations, it generates a simple, standard data format, thereby **emulating** the style of simpler environments. Our approach diverges from Gym, PettingZoo, and Supersuit in three significant ways:

1. PufferLib consists of a single wrapper layer with transformations applied in a fixed sequence. Observations are grouped, then featurized, subsequently flattened, and finally padded and sorted.

2. It provides utilities for both flattening and unflattening observations and actions without the issues described earlier.

3. The wrapper class is procedurally generated using data scoped from a dummy instance of the unwrapped environment, enabling the static precomputation of a few costly operations.

The emulation layer starts with a Binding object. Users can instantiate a binding from a Gym or PettingZoo environment class, instance, or creation function. They can supply several arguments to the Binding object, including a custom postprocessor for features, actions, and rewards, choices about flattening observation and action spaces, whether to pad to a constant number of agents, whether to truncate environment simulation at a set number of steps, etc. The Binding class creates or uses the provided environment instance and resets it to yield an initial observation. This observation, alongside the provided binding arguments, is used to create a wrapper class for the environment. The significance of this process is that it allows the initial observation to be statically scoped into the wrapper class. The Binding then offers access to the wrapper class with no intermediate layer.

The wrapper class is designed to address all the common difficulties associated with working with complex, multi-agent environments as simply as possible. For context, it totals only around 800 lines of code, which further shrinks excluding the various API usage, input checking exceptions, optional correctness checks, and utility functions. By comparison, the core of PufferLib is shorter than the domain-specific code previously used to support Neural MMO alone. In an ideal world, users would never face uncaught errors in internal libraries. However, as no reinforcement learning library to

date has achieved this standard, PufferLib provides a pragmatic solution by offering a simple, single source of failure, as opposed to the potential confusion caused by dozens of conflicting wrappers.

## 4.2 Vectorization

Existing vectorization tools build into Gym and PettingZoo lack stable support for multi-agent environments. PufferLib bridges this gap by including a suite of three vectorization tools. These tools leverage the sanitized output format provided by the emulation layer, allowing them to be both performant and simple. Each environment will consistently present the same number of agents, in the same order, with flattened observations. The three vectorization backends are as follows:

1. **Multiprocessing:** This tool simulates $n$ environments on each of $m$ processes, totaling $nm$ environments, using Python's native multiprocessing. An additional version, which transfers observations via shared memory, is included. This variant can prove useful for environments demanding high data bandwidth.

2. **Ray:** This tool, like the multiprocessing one, simulates $n$ environments on each of $m$ processes, using Anyscale's Ray distributed backend. Although this implementation might be slower for fast environments, it works natively on multi-machine configurations. It also includes a version that transfers observations to the shared Ray memory store instead of directly to processes, which can be faster for specific environment configurations.

3. **Serial:** This tool simulates all of the environments on a single thread. This setup proves useful for debugging, as it is compatible with breakpoints while maintaining the same API as the previous implementations. Additionally, it is faster for extremely low-latency environments where the overhead of multiprocessing outweighs its benefits.

All these backends offer both synchronous and asynchronous APIs, facilitating their use in a buffered setup. In this configuration, the model processes observations for one set of environments while another set of environments processes the previous set of actions. Additionally, all these backends provide hooks for users to shuttle any arbitrary picklable data to the environments. This feature is essential for advanced training methods that need to communicate - for instance, new tasks or maps - with specific environments on remote processes.

## 4.3 Integrations

The current release of PufferLib includes support for CleanRL and RLlib, with an extension to Stable Baselines [Raffin et al., 2021] projected for the forthcoming minor versions. Owing to the consistent and standard format defined by the emulation layer, even for complex environments, it is relatively straightforward to employ the same PyTorch network across different framework APIs. PufferLib introduces a PyTorch base class that separates the *forward()* function into two parts: *encode_observations* and *decode_actions*. Functions preceding a recurrent cell are categorized under the encoding function, and those succeeding it are under the decoding function. This division is implemented because the handling of recurrence is often the most challenging difference among various frameworks. In addition, the mishandling of data reshaping in the recurrent cell is a common source of implementation bugs. We provide additional checks to mitigate this risk. On top of this API, PufferLib constructs a small, per-framework wrapper, which activates the user-specified recurrent cell according to the specific requirements of the given framework. This approach may be expanded to include transformers in the future, although most RL frameworks currently lack support for this.

## 5 Materials Available for Release

The public version of PufferLib (version 0.3) is accessible at pufferai.github.io. Version 0.4 is planned for release by the end of the summer and will include additional framework support. User testing greatly accelerates progress, and the exposure from publication would significantly benefit this work. We currently have the following materials ready for release:

- Simple documentation and demos for CleanRL and RLlib with Neural MMO available on the website mentioned above.

- Built-in support and testing for Atari Bellemare et al. [2012], Butterfly (part of PettingZoo), Classic Control (part of Gym), Crafter Hafner [2021], MAgent Zheng et al. [2017], MicroRTS [Huang et al., 2021b], Nethack [Küttler et al., 2020], Neural MMO [Suarez et al., 2021], and SMAC [Samvelyan et al., 2019] with partial support for Griddly [Bamford et al., 2020] and planned extensions to DM Lab [Beattie et al., 2016], DM Control [Tassa et al., 2018], ProcGen [Cobbe et al., 2019], and MineRL [Guss et al., 2019]. Most of these are one-line bindings that primarily depend on ensuring compatibility of dependency versions. These are also included in our correctness tests.

- A Docker container, fondly referred to as PufferTank, that comes pre-built with PufferLib and all of the above environments pre-installed.

- Baselines on the 6 original Atari environments from DQN [Mnih et al., 2013], sanity-checked against CleanRL's vanilla implementation.

- A community Discord server with 100 members, offering easy access to support.

This version further includes an advanced set of correctness tests that reconstruct the original environment data format from the final version postprocessed by PufferLib. This has aided us in identifying several dozen minor bugs in our development builds. PufferLib is also being utilized in the upcoming Neural MMO competition, enabling much simpler baseline code than would be achievable without it.

## 6 Limitations

The most significant limitations of the current release of PufferLib include

1. No support for heterogenous observation and action spaces. These are difficult to process efficiently in a vectorized manner.

2. No support for continuous action spaces. This may be supported with a medium amount of development effort in future versions.

3. Environments must define a maximum number of agents that fits in memory. Additionally, agents may not respawn. The former is a fundamental limitation of the underlying PettingZoo binding. The latter may be supported in a future version with a small amount of development effort.

Additionally, as the first publication release of a new framework, we are heavily reliant upon growing a user base to ensure the stability of our tools. We run a battery of correctness tests and verify training performance on Atari in each new release, but subtle bugs have occasionally slipped through during development.

## 7 Conclusion

This paper introduces PufferLib, a versatile tool that greatly simplifies working with both single and multi-agent reinforcement learning environments. By providing a consistent data format and handling complex transformations, PufferLib allows researchers to focus on model and algorithm design rather than the quirks of their environments. Its built-in support for a wide variety of environments, coupled with its scalability and compatibility with popular RL frameworks, makes PufferLib a comprehensive solution for reinforcement learning tasks. We welcome the open-source community to use and contribute to PufferLib, and we anticipate that its ongoing development and integration will continue to lower barriers in reinforcement learning research.

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
