# Supplementary Material for PufferLib

Joseph Suarez

June 7, 2023

## 1   Documentation and Intended Uses

Documentation is available at pufferai.github.io. PufferLib is intended to be used as a wrapper layer that sits between simulated environments and reinforcement learning libraries. Its purpose is to enable compatibility between complex simulated environments and simple, easy-to-use reinforcement learning libraries that would otherwise not be usable together. The intent of this work is to push research towards more cognitively interesting training tasks by default by simplifying the process of conducting such work.

## 2   Author Statement

The author bears all responsibility in case of violation of rights etc, though none such are applicable because PufferLib is purely a code + docs release with no user data involved. The data license for everything but logo assets (i.e. the PufferLib pufferfish) is MIT. Author retains copyright to the logo.

## 3   Hosting, Licensing, and Maintenance plan

Code and docs hosted by GitHub, anticipated updates to be licensed under MIT or similarly permissive terms. Development will continue for at least the next year with support and maintenance orchestrated via our community Discord (discord.gg/puffer) during and thereafter.