# OpenReview forum: "PufferLib: Making Reinforcement Learning Libraries and Environments Play Nice"
_NeurIPS.cc/2023/Track/Datasets_and_Benchmarks — Submitted to NeurIPS 2023 Datasets and Benchmarks_

### Official Review · Reviewer_C2b5 · 2023-07-06
**A Useful Tool with Minor Limitations**

**Rating:** 6
**Confidence:** 2
**Correctness:** Yes.
**Clarity:** The _Problem Statement_ section in un…

**Strengths:**

See above.

**Additional Feedback:**

# References

[1] Andrew Patterson, Samuel Neumann, Martha White, Adam White. Empirical Design in Reinforcement Learning. 2023.

[2] Jiayi Weng, Min Lin, Shengyi Huang, Bo Liu, Denys Makoviichuk, Viktor Makoviychuk, Zichen Liu, Yufan Song, Ting Luo, Yukun Jiang, Zhongwen Xu, Shuicheng Yan. EnvPool: A Highly Parallel Reinforcement Learning Environment Execution Engine. 2022.

[3] Logan Engstrom, Andrew Ilyas, Shibani Santurkar, Dimitris Tsipras, Firdaus Janoos, Larry Rudolph, Aleksander Madry. Implementation Matters in Deep RL: A Case Study on PPO and TRPO. 2020.

[4] Peter Henderson, Riashat Islam, Philip Bachman, Joelle Pineau, Doina Precup, David Meger. Deep Reinforcement Learning that Matters. 2018.

**Documentation:**

The examples are much too complex, and no minimal working examples exist (see above).

**Limitations:**

# Limitations

## Inability to use Custom Algorithms and Environments Reduces the Usefulness of PufferLib

Whether the user can easily add their own environment or algorithm implementations to PufferLib is unclear. Much research on algorithm design revolves around the researcher implementing their own algorithm and testing that algorithm on a suite of environments. Furthermore, it is oftentimes useful to create a custom environment to test a specific problem. Can users of PufferLib add their own algorithm and environment implementations to the system easily? If not this seems to be a severe limitation.

## Multiprocessing in Python is Slow

One limitation of PufferLib is that it is written in Python, which is known to be slow and have poorer multiprocessing than other languages. Does PufferLib introduce a performance degradation for individual environments, especially since so many operations must be performed in the background? I would think not, since whether or not PufferLib is used, those data transformations must still take place (although I would imagine PufferLib adds extra operations behind-the-scenes for housekeeping tasks).

What about for multi-environment experiments? Does PufferLib introduce a significant overhead when using multiprocessing with multiple environments? I would imagine that it does run slower than other frameworks like EnvPool [2] which are implemented in C++. How significant would this overhead be? Would the extended runtime (compared to a framework like EnvPool) be worth the conveniences that PufferLib brings, or would it be better to use a more performant multi-environment multiprocessing framework? This is especially important considering that not all researchers have access to clusters or powerful computing devices to run their experiments on. Is it possible to utilize frameworks like EnvPool to speed up the runtime of PufferLib?


**Opportunities For Improvement:**

# Opportunities for Improvement

## Documentation
Documentation at some points is quite complex and could be simplified. In the examples section on the website, the _minimal_ examples are too complicated. It may be good to have multiple levels of examples, since the examples should demonstrate *how to use* PufferLib, and not how to create/manage agents in different frameworks, use WandB, etc. A good, minimal example would be:
```
agent = create_agent()
env = create_env()
# Actual loop performing the agent-environment interaction
```

I get completely lost in the current examples because too much is going on.

## Spelling in Figure 1

In figure 1, one of the pink boxes reads
> PufferLib emulation hook inside user PyTorch model unpack flat observations into their original structured format

which I think should read

> PufferLib emulation hook**s** inside of the user**\'s** PyTorch model **to** unpack flat observations into their original structure format

## Literature Review on Implementation

One benefit of PufferLib is that most data processing is performed in a single codebase, which can limit the number of introduced bugs in algorithm/environment implementations. Futhermore, having a single implementation of data processing can reduce the number of confounding factors when comparing multiple algorithms. The paper could be improved by discussing how implementations can affect conclusions drawn in research projects, and how PufferLib could circumvent these issues. Some useful references are [1], [3], and [4].

## Unclear Problem Statement Reduces the Perceived Need for PufferLib

The problem statement is unclear. The section is hard to follow, which reduces the perceived need for PufferLib. After reading the section, it was not immediately clear that PufferLib is needed. Doesn't PufferLib just move all these computations elsewhere? One suggestion to make this section more clear is to have a figure/flowchart of operations and reference parts of that figure in the text.

**Relation To Prior Work:**

Yes.

**Summary And Contributions:**

# Summary
PufferLib is a tool to standardize the processing of data in reinforcement learning environments, particularly useful for multi-agent, parallel environments. PufferLib has wrappers for many different reinforcement learning frameworks (environments and learning algorithms) and allows algorithm implementations from different codebases to seamlessly integrate with environments from other codebases.

# Main Argument

Using a standard tool such as PufferLib for data processing in reinforcement learning environments has a number of benefits, some of which are:

1. By using a standard tool for data processing, many bugs can be localized to a single framework, making debugging easier and limiting the number of bugs in code.
2. By having a tool for data processing, the creation of new multi-agent reinforcement learning environments can be streamlined.
3. By using a standard tool for data processing, more time can be devoted to algorithm development
4. With many people contributing to a standard open-source data processing tool such as PufferLib, bugs/features will be caught/implemented quickly

Currently, each individual environment framework processes complex, hierarchical environment data itself. As outlined in the paper, this has led to a plethora of bugs in current frameworks. Such implementation-level bugs can affect the conclusions of researchers and has been shown to have potentially affected the conclusions drawn in popular, highly impactful papers [1].

By offering a standard data processing solution, PufferLib makes a significant contribution to the improvement of existing and new reinforcement learning frameworks and benchmarks. Low-level and implementation-level details can remain contained in a single framework, localizing bugs and performance issues, and allowing researchers to focus on algorithm development and understanding rather than on implementations. Unfortunately, it is not clear whether users of PufferLib can easily add their own algorithms or environments to the system, and PufferLib may only be useful when experimenting with existing algorithms and environments in the few supported codebases.

---

> ### Author Response · Authors · 2023-08-09
> **One Line Environment Support + Documentation and Impact Clarifications**
>
> Thank you for your thorough review of my work. You will be happy to hear that new environments can typically be supported with a single line of code. This is what it will look like once I merge the latest changes:
>
> For single agent environments:
>
> env = pufferlib.emulation.GymPufferEnv(env=env)
>
> For multi agent environments:
>
> env = pufferlib.emulation.PettingZooPufferEnv(env_creator=your_env_creator, env_args=args, env_kwargs=kwargs)
>
> The API will support adding environments by the environment class, a make_env function, or by wrapping an environment object. You can use the PufferLib emulation layer to make complex environments easier to work with in any code base. For example, if you want to write a new algorithm, PufferLib allows you to assume that your data storage will be flat, fixed-size tensors. This significantly simplifies implementations. If you would like to work with a vectorized interface, PufferLib provides one out of the box. This is typically non-trivial for complex or multi-agent environments.
>
> Documentation: You can use PufferLib environments identically to standard PettingZoo environments. I will add a basic example of this, but it is the same as here: https://pettingzoo.farama.org/api/parallel/. The minimal example is long because it is inherited from CleanRL, a widely popular RL framework. CleanRL does not work like most libraries in that you do not import it. Instead, you copy the single-file algorithm implementation that you want and modify it directly. The point of this example was that you can use PufferLib in CleanRL by modifying only a few lines of code. I will include a git diff or similar in the docs to clarify this.
>
> Problem Statement: The key innovation of PufferLib is to move per-environment data processing into a single library. Before integrating PufferLib, Neural MMO had hundreds of lines of buggy single-purpose data transformation code. Now, it has a one-line PufferLib binding that is tested for correctness on a dozen different environments. I agree that the flowchart could be clarified. Do you have any suggestions for how to clean this up?
>
> Multiprocessing: Yes, this is a problem, but there are no other solutions at the moment. EnvPool is targeted towards simpler environments and does not support any of the key features that PufferLib does: complex observations and actions, multi-agent environments, etc. EnvPool also does not work with C/C++ environments that require Python bindings, such as NetHack. PufferLib is targeted instead towards complex environments where, currently, it is difficult to run them at all. Personally, I am betting that RL will begin using larger networks, which would enable you to saturate your GPUs with Python environments on only a few cores.
>
> Minor: Thank you for the additional references. I will also fix the typos that you have pointed out.
>
> If the above clarifications improve your evaluation of my work, kindly consider increasing your score by a point. Or, if you prefer, feel free to request additional features or edits.

---

> > ### Comment · Reviewer_C2b5 · 2023-08-14
> > **Response**
> >
> > Thank you for your response. I would just like to clarify my comment about the "Problem Statement" section.
> >
> > Specifically, lines 52 - 79 refer to many different operations that happen in code. I think that the explanations here are difficult to understand, especially for users who have not used frameworks such as PettingZoo. My suggestion to clarify this section is the following. You could have a new figure/flowchart of each operation that is performed, starting with the environment initialization and configuration. In the text, you could then refer to each section of this flowchart.

---

> > > ### Author Response · Authors · 2023-08-18
> > > **Clarification of Internals**
> > >
> > > I will improve the presentation of this material in the paper. Do note that, if your concern is for users of the library, these are ultimately internal implementation details of PufferLib. I present them in the paper to highlight why this seemingly simple engineering problem is actually quite tricky, as well as to show where other libraries have gotten it wrong.
> > >
> > > From the user perspective, PufferLib simply provides wrappers that make environments work with learning libraries that would be otherwise incompatible. See the general comment at the top for the latest docs, which include a much more thorough walkthrough of typical usage.
> > >
> > > Please let me know if there are any other improvements I could make that would merit an increase to your score.

---

### Official Review · Reviewer_J5Gf · 2023-07-21
**Great concept and tool. Paper lacks impact results.**

**Rating:** 7
**Confidence:** 3
**Correctness:** Yes.
**Clarity:** Yes.

**Strengths:**

+ Solves an important problem
+ Well documented
+ Extensible

**Additional Feedback:**

None.

**Documentation:**

Yes.

**Ethics:**

No.

**Limitations:**

Clearly mentioned.

**Opportunities For Improvement:**

See summary above.

**Relation To Prior Work:**

Yes.

**Summary And Contributions:**

PufferLib tries to address the challenging task of unifying the growing number of RL environments to create standardized, self-contained workflows for training RL agents. This is crucial if we are going to be able to fairly benchmark and evaluate algorithms (e..g, PPO vs. SAC) and computer hardware (e.g., GPU vs. TPU) across a diverse range of applications and contexts. Pufferlib also wraps some of the most important environments in RL today. PufferLib also comes with really good documentation and appears to be quite extensible.

The main challenge the reviewer has with this paper is the lack of results. Clearly the library is being used and the one case study described seems to indicate that it is useful. However, there are no graphs or charts or even qualitative analysis at scale to support the claims made by the author and to support the importance and early impact of the library. As such, it appears to be a nice idea that may or may not actually be useful in practice (although the reviewer notes that it is most likely very useful in practice -- it is just that the paper itself doesn't make that abundantly clear outside of claims). Some code snippets showing how easy it is to add new environments or something along those lines might also help.

In short, the library seems great but the paper feels like a work in progress and not a paper with results.

---

> ### Author Response · Authors · 2023-08-09
> **One Line Environment Support + Clarification of Impact**
>
> Thank you for your time in reviewing my work. It looks like I buried some of the information you were looking for in the documentation on the long examples page. This is how you currently add a new environment:
>
> binding = pufferlib.emulation.Binding(env_creator=make_env,
>             default_args=[env_name, framestack],
>             env_name=env_name,
>             postprocessor_cls=AtariFeaturizer,
>             emulate_flat_atn=True,
>             record_episode_statistics=False,
> )
>
> After the current refactor is merged, it will be even easier:
>
> env = pufferlib.emulation.GymPufferEnv(env=env)
>
> Instead of relying on a custom Binding API, the new implementation creates a familiar env object. As a multi-agent example, this is the entire wrapper for Neural MMO:
>
> env = pufferlib.emulation.PettingZooPufferEnv(env_creator=nmmo.Env, env_args=args, env_kwargs=kwargs)
>
> I agree that I should have better data to back the importance of the library. This has been difficult because there is often no point for comparison. For example, the alternative to using PufferLib for Neural MMO is to just not use Neural MMO. It is a standard PettingZoo environment, but no RL library correctly supports it. By comparison, with PufferLib, Neural MMO works with CleanRL out of the box. Every other RL environment with a correct Gym or PettingZoo API should work with PufferLib with a 1 line wrapper. I am also working on support for popular environments that have errors in API conformity.
>
> What would you like to see in order to increase your score by a point? I would be more than happy to make edits to the paper or add additional environments during the discussion period.

---

> > ### Comment · Reviewer_J5Gf · 2023-08-09
> > **Thank You for Your Response**
> >
> > Thank you for you response. I think if you make it more clear that with one line under the new refactoring you can add any new environment with a PettingZoo or Gym API, and make it clear that the alternative is to not be able to use popular environments such as Neural MMO, that should satisfy many of my concerns. One thing you could also consider is placing a table of popular environments (and possibly their github stars as a metric) that could be easily used through PufferLib in this way, that could make the impact of PufferLib even more clear.
> >
> > Finally, addressing the concerns of other reviewers, many of which you have already suggested good ways to address from my reading of your comments below, would further alleviate many of my concerns.

---

> > > ### Author Response · Authors · 2023-08-18
> > > **Updated as requested**
> > >
> > > I have incorporated your suggestions into the library and docs, including example code and star count -- see the general comment at the top of this page. If you feel this is a material improvement to the library, please consider increasing your score. If not, kindly let me know what other changes you would like to see.

---

### Official Review · Reviewer_gJrA · 2023-07-21
**A useful tool to bridge the gap between reinforcement learning libraries and control environments**

**Rating:** 6
**Confidence:** 3
**Correctness:** To the best of my knowledge, the clai…

**Strengths:**

* Pufferlib handles all the wrappers and interacting with the environment under the hood, so that researchers no longer need to debug in source code of environment libraries like [Gym](https://github.com/openai/gym), and can enjoy an "emulator" experience.
* Pufferlib provides integrated flattening and unflattening of data so structured data can be handled easily.
* Pufferlib builds in three strong vectorization tools: Python's native multiprocessing, [Anyscale Ray](https://github.com/ray-project/ray), and a serial vectorization tool for debugging.
* Pufferlib introduces a base class for PyTorch networks which separates the `forward()` function into two parts to help handle recurrence.

**Additional Feedback:**

Overall, I'm happy to see a tool that can solve the pain of digging into `gym`'s source code to find bugs. The library is still in a early period and I look forward to seeing the author as well as the open source community continue to improve it.

As of now, a minor issue would be better dependency management. While installing PufferLib, I encountered the following message:
```
ERROR: pip's dependency resolver does not currently take into account all the packages that are installed. This behaviour is the source of the following dependency conflicts.
wandb 0.12.21 requires PyYAML, which is not installed.
tensorflow-probability 0.20.1 requires decorator, which is not installed.
tensorflow-probability 0.20.1 requires gast>=0.3.2, which is not installed.
jaxlib 0.4.1+cuda11.cudnn86 requires scipy>=1.5, which is not installed.
jax 0.4.1 requires scipy>=1.5, which is not installed.
flax 0.6.1 requires PyYAML>=5.4.1, which is not installed.
dm-control 1.0.14 requires lxml, which is not installed.
dm-control 1.0.14 requires scipy, which is not installed.
```
I then installed these required packages manually.

**Clarity:**

In general, the paper is clearly written, but there are a few suggestions:
1. On L194 of the paper, the link to PufferLib could be wrapped in the `\url{}` format.
2. On lines 200 and 201, the formats of 3 citations are incorrect; should be "[Author, Year]" instead of "Author, [Year]" unless Author is being intentionally addressed.

**Documentation:**

I found the installation to be relatively easy as PufferLib is already available to install with `pip`. However, the demo code is quite long, and I would like to see better documented code to go through the features of the library in a more step-by-step manner.


**Ethics:**

I do not see any ethical concerns in this work.


**Limitations:**

The authors address the limitations of PufferLib adequately - no support for heterogenous observation and action spaces, continuous action spaces, and environments must define a maximum number of agents that fits in memory.

The current largest limitation of PufferLib in my opinion is not supporting continuous action spaces.

**Opportunities For Improvement:**

* The [Minimal CleanRL demo code](https://github.com/PufferAI/PufferLib/blob/0.3/cleanrl_minimal.py) contains a minor bug in L248, where I can only write one element of the array to Tensorboard.
* [Gym](https://github.com/openai/gym) is already deprecated and replaced by [Gymnasium](https://github.com/Farama-Foundation/Gymnasium). The authors should consider updating to [Gymnasium](https://github.com/Farama-Foundation/Gymnasium), as [Gym](https://github.com/openai/gym) is no longer maintained.

**Relation To Prior Work:**

PufferLib builds on [Gym](https://github.com/openai/gym) and [PettingZoo](https://github.com/Farama-Foundation/PettingZoo) but also addresses their limitations, specifically solving the issue of using multiple layers of wrappers, wrong multi-agent key ordering, and improving vectorization with more standard libraries.


**Summary And Contributions:**

This paper introduces PufferLib, a middleware solution for RL to address the limitations in complex environments. By converting intricate environments into a standardized, vectorized format, PufferLib eliminates the need for time-consuming conversion layers, simplifying RL implementation. It offers bindings for easier user interaction and is compatible with some of the popular RL frameworks, making it a versatile tool for both single and multi-agent RL tasks.

---

> ### Author Response · Authors · 2023-08-09
> **Gymnasium Support + Continuous Actions Coming Soon!**
>
> Thank you for your detailed commentary. Yes, I plan to include support for Gymnasium, but this is unfortunately not as simple as just updating. Many widely used environments still use the old API, so PufferLib will need to support both. Doing this cleanly requires significant refactoring of the code base, which I am almost finished with. I am also including functionality to enable the PufferLib tests to run on environments with conflicting Gym versions as part of this update. This should be done by the end of the summer.
>
> Supporting continuous action spaces is a little bit trickier. I am aware that this is something many folks would be interested in. This feature mainly involves some difficult changes to the way I flatten and unflatten spaces. If you would be so kind as to commit to a 1 point increase to your score upon the inclusion of this feature, I will break out the midnight oil and try to merge it during the review period.
>
> Minor: The demo code bug looks inherited from the upstream CleanRL PPO implementation. My goal was to demonstrate use of PufferLib with as few modifications to the original script as possible. If you PR a fix to CleanRL, I’ll make the corresponding change to the demo code. I will fix minor typos as well, thank you for catching these.
>
> Documentation: The minimal demo is built on CleanRL, one of the most widely used RL libraries. It is not meant to be installed like a normal library, but copied and modified directly. Perhaps I could better comment on the changes for using PufferLib or link a git diff? Happy to include other specific tutorials if you have suggestions.
>
> Dependencies: PufferLib itself does not have any tricky to install dependencies. These come from the included environments. To help with this, I am also developing PufferTank https://github.com/PufferAI/PufferTank, a container prebuilt with all of the dependencies. Pip package errors will be corrected.

---

### Official Review · Reviewer_CCab · 2023-07-21
**Practical Contribution**

**Rating:** 6
**Confidence:** 3

**Strengths:**

1. This seems like the sort of library that, if maintained and popularized, may make RL more accessible. Particularly since RL is not sample-efficient, and improving robustness/ease of use for improved parallelization could potentially make the lives of many researchers (e.g. with small-ish academic clusters) easier.

2. While the wrapper pattern that is currently popular has its advantages (composability to fit your use-case), I have encountered my share of bugs within them, and I tend to think they a little over-used and are not suitable for all scenarios. Standardization around data-parallelism-related functions makes sense to me.

**Additional Feedback:**

I appreciate when API documentation has convenient links from the methods documented to their location in source code. This is not a concern, but I rarely get an opportunity to mention this directly to library developers. I'm also uncertain how difficult it is, as I have not done it myself.

**Clarity:**

Overall it is quite clear, with my main exception being the confusion I had in the seeming lack of explanation of how PufferLib is handling recurrence in Section 4.3.

**Correctness:**

I am unsure how to evaluate this for this work. This paper is neither a benchmark nor a dataset, but rather a tool that makes existing benchmarks more usable for others. The value it offers is in improving the experience of other researchers, and while I believe this is worthwhile, I am unsure how to thoroughly analyze it for soundness along this axis, short of installing the library and trying it myself. The work seems to have been validated, in a sense, via how it improved the function (discovered bugs, simplified code) of an existing benchmark, but on this axis I am unsure how this would be done with statistical rigor.

On the other hand, as I mentioned in Opportunities for Improvement, this work has not been analyzed for speed or performance on any of the benchmarks it wraps. As such a library would be expected to not be a huge hit to speed, and to be comparable on performance results, I think such results would increase my ability to say that this work is correct.

**Documentation:**

The library's API is documented, including examples. At a glance it looks mostly like what I would expect documentation for such a library to look like. I had little difficulty using it to answer a few questions I had about the paper.

**Ethics:**

I do not suspect there are ethical concerns.

**Limitations:**

I believe I covered most of the limitations that occurred to me in the Opportunities for Improvement, coupling them with potential suggestions for thoughts on improvement.

**Opportunities For Improvement:**

I poked around the code repository a little, but did not have the time to extensively. Some of my recommendations for improvement may already be addressed, and if so I apologize.

1. I have used three different forms of asynchronous RL over time:
    * Parallel environment stepping, gather observations, model run as a batch, re-distribute out to parallel environments (as in A3C). I believe this is what is usually meant by "vectorized environments".
    * Full parallelization of actors, such that each process also handles running the forward model, but for algorithmic reasons training cannot be effectively parallelized. (This might be kind of a niche use case.)
    * Full parallelization of both actors and learners (IMPALA), where separate actor processes constantly populate a buffer, and the learner grabs any filled buffers and trains on them.

    I have switched pretty much exclusively to using the last one, as it has been vastly more performant for me, but I am sometimes inclined to experiment differing patterns of collection/training. Based on the description of parallelization in Section 4.2, it seems as though PufferLib only supports the first one. I believe Ray supports the last one, but I have not used it so I am not sure.

    Which brings me to my question/source of uncertainty: how extensible is PufferLib (by you for common cases, or by your users for less-common use cases) to changing the pattern of data-flow? Is it extensible enough as-is that I could simply specify that I wish to use Ray's IMPALA, and everything would work under the hood? If such use-cases have not been considered, I would strongly encourage their consideration.

1. Changing the PyTorch model standard pattern (from forward() to encode_observations() and decode_actions() seems like a significant change, at least in the sense that, if this new pattern is non-optional, the significant number of policies that already exist cannot be used out of the box with this library. Is it possible to offer this PufferLib-managed recurrence as optional, but still being able to use the standard forward() API (with the acknowledgement that you won't get the offered recurrence-management-safety features)?

1. Related to #2, the description of what exactly PufferLib is doing to enable recurrence is lacking. In Section 4.3, it is described that encode is before the recurrence and decode is after, but it does not address how the recurrence specified, what assumptions it makes, nor what flexibility it allows.

    If I wanted a method with (embedding -> LSTM -> <some other network> -> LSTM), I do not think I would be readily able to use the existing method to do it. This seems to me like imposing structure on the problem that will limit the accessibility of your method to others. (Also minor: "decode_actions" was to me, a misleading name. My instinct was that it was some sort of processing over already-computed actions, not the place where an embedding is turned into an action.)

Overall, I realize that there is a trade-off between imposing structure to realize the gains of standardization (e.g. robustness) and allowing flexibility to capture a wider variety of use-cases. It seems likely that the balance struck in PufferLib will be of use, but it seems like some of the features being offered would be very widely useful, and I am worried that increased structure to improve more narrow use-cases will limit its adoption.

I would also like to suggest that the library be evaluated for speed, by running it versus the libraries it wraps using their standard method of being invoked. It would be useful to see whether any overhead is being incurred. It could also be useful to simply give results during training, and show that they are within error of prior methods, to validate that the library is operating as intended.

**Relation To Prior Work:**

Yes, there is a clear comparison with existing work. I am unsure if the related work is exhaustive, however. I believe this is the first time I've seen a Related Works section completely devoid of references (the works it mentions, it referenced earlier in the paper).

I think the authors do a reasonable job of justifying their work in relation to others, however, including concrete examples of problems that other methods have difficulty solving.

**Summary And Contributions:**

This paper presents a library, PufferLib, designed to help bridge the standardization gap between a number of popular reinforcement learning libraries, which currently presents a source of bugs and hassle, particularly with regard to multiprocessing/multi-machine scenarios. There seems to have been a particular motivating force particularly surrounding more complex multi-agent scenarios.

---

> ### Author Response · Authors · 2023-08-09
> **Broadly Compatible Emulation + Native PyTorch Networks -- Enough for Accept?**
>
> Thank you for your extensive feedback and for even looking through my code! I am currently in the process of merging some major refactors to the library that will make the core a lot easier to follow and test, as well as improve performance a bit.
>
> PufferLib’s emulation layer should work with any framework. All it does is make complex environments conform to a restricted, simpler set of the Gym/PettingZoo API without loss of generality. This is important because most learning frameworks get the details of handling the full API wrong. For example, last I checked, RLlib’s Impala does not work with Neural MMO out of the box, but it does work if you wrap it with PufferLib. Outside of the stated limitations, it’s one line to wrap any new Gym or PettingZoo environment. Try it out, and if it doesn’t work, I’ll personally help you debug it and patch PufferLib accordingly. This offer is open to everyone on the PufferLib Discord as well.
>
> The vectorization tools are limited to libraries that support vecenv interfaces. The nice thing about these is that they make environments as complex as Neural MMO work with CleanRL, arguably the simplest library out there.
>
> You’re spot on with the PyTorch model change. I should have thought of that and will make the encode/decode API optional. The only reason I did this in the first place was that, in my experience, these few simple checks on recurrence catch 80% of tensor shape bugs. This should satisfy your third point as well.
>
> PufferLib adds virtually no additional structure to familiar workflows. Envs still behave like gym/pettingzoo envs, the vectorization layer behaves like every other vectorization tool, and (with the above change) models are just pytorch models.
>
> I am working on more comprehensive benchmarks and performance tests. Right now, I have verified training roughly matches CleanRL on Atari with 5% overhead, and the Neural MMO competition has a working baseline with PufferLib. For complex environments like Neural MMO, overhead is more significant, but there is no good way to use the environment without PufferLib anyways. The main overhead per environment comes from flattening and unflattening observations. VecEnv does not scale well, but that is a Python multiprocessing limitation. We also have a Ray backend for vectorization.
>
> Minor fixes: I can move citations to related works or duplicate them if you prefer. I will also clarify the language around the recurrence wrapper. It’s much simpler in my latest dev branch.
>
> If the above context + PyTorch change sways you, please consider increasing your score. This project would benefit tremendously from the user testing that publication exposure would provide.

---

> > ### Comment · Reviewer_CCab · 2023-08-31
> >
> > Thank you for addressing my comments. I certainly appreciate the utility of such a method, and have raised my score from a 5 to a 6.
> >
> > I don't think duplicating the citations in the related works is necessary, I guess I just found it a little notable that only more detail needed to be provided on already-cited works, and no new works were relevant.

---

### Author Response · Authors · 2023-08-18
**Updates to PufferLib**

Thank you once again to all reviewers. I have made improvements to the library based on your suggestions and can share a preliminary build of the the new library.

New documentation: https://pufferai.github.io/dev
Current dev branch: https://github.com/PufferAI/PufferLib/tree/experimental

Here are some highlights:

1. **Compatibility with any and all learning libraries.** Our wrappers no longer create intermediate binding objects or convert Gym envs to PettingZoo envs. If you give PufferLib a Gym env, you get a Gym env back. Same for PettingZoo. This will work with any environment outside of stated limitations (I am working on continuous actions + gymnasium support). Here's how to handle NetHack and Neural MMO, two of the most notoriously difficult environments to work with:

```
import pufferlib.emulation
import nle
import nmmo

nethack = pufferlib.emulation.GymPufferEnv(env_creator=nle.env.NLE)
neural_mmo = pufferlib.emulation.PettingZooPufferEnv(env_creator=nmmo.Env)
```
2. **PyTorch API is now optional.** You can use our easy framework integrations with vanilla PyTorch policies, without having to split your forward function into encode and decode portions.

3. **~100x faster flatten and unflatten.** This was the only major source of overhead in PufferLib. It is now a C extension and, to my knowledge, the fastest flatten/unflatten implementation available for nested Python data.

4. **Much higher code quality and test coverage.** This includes performance and regression tests for each emulation and vectorization module.

The code samples in the new documentation require the latest version of PufferLib and have a few small bugs, but they will be updated and available as a runnable Colab notebook soon.

---

> ### Author Response · Authors · 2023-08-26
> **Run PufferLib in your Browser!**
>
> I have added two demos to the dev website pufferai.github.io/dev. Training has been tested against base CleanRL with Atari. I just need to make sure performance matches or exceeds 0.3 on Neural MMO and then I can ship it as 0.4 and release pip packages.
>
> **PufferLib API Demo:** https://colab.research.google.com/drive/1l1qLjerLwYoLjuKNr9iVc3TZ8gW2QVnz?usp=sharing
>
> This demo walks users through all of the core functionality of PufferLib. It uses Neural MMO and NetHack, two environments that are notoriously difficult to work with. Less than 15 lines of code to run either distributed, including computing actions with a real model.
>
> **Train with CleanRL:** https://colab.research.google.com/drive/1OMcaJnCAF1UiCJxKIxSS-RdZTuonItYT?usp=sharing
>
> Use PufferLib with CleanRL by changing ~5 lines of code. I use Atari for the demo because most researchers are familiar with it, but you can swap in any Gym/PettingZoo environment (subject to a few temporary limitations mentioned in the paper). 1-3 lines for this + your new model.
>
> I also have two more scripts are linked on the website: a small RLlib integration and CleanPuffeRL. The RLlib integration is a bit tricky because the developers pinned a very recent version of Gymnasium that is not compatible with most common research environments, and older RLlib builds are not stable. CleanPuffeRL is my heavily modded version of CleanRL with advanced features like async sampling, self-play, and custom postprocessors. These will be added as Colab demos once they are more stable.
>
> Please note that the demos clone the dev branch and I am still making final commits.

---

### Decision · Program_Chairs · 2023-09-22

**Decision:**

Reject

**Comment:**

This work provides a software library and middleware for standardizing reinforcement learning environments. After the discussion period, the reviewers and I are in agreement that this software contribution is not sufficient for publication in the datasets and benchmark track, as it does not provide a new dataset or benchmark directly, and the software contribution does not lead to any significant methodological improvement or insight. We recognize the value in research software and middlelayers and encourage the author to continue developing the library and consider a venue focused more on software contributions, such as JMLR's MLOSS.